# Laser Synthesized Core-Satellite Fe-Au Nanoparticles for Multimodal In Vivo Imaging and In Vitro Photothermal Therapy

**DOI:** 10.3390/pharmaceutics14050994

**Published:** 2022-05-05

**Authors:** Olga Yu. Griaznova, Iaroslav B. Belyaev, Anna S. Sogomonyan, Ivan V. Zelepukin, Gleb V. Tikhonowski, Anton A. Popov, Aleksei S. Komlev, Petr I. Nikitin, Dmitry A. Gorin, Andrei V. Kabashin, Sergey M. Deyev

**Affiliations:** 1Shemyakin-Ovchinnikov Institute of Bioorganic Chemistry, Russian Academy of Sciences, Moscow 117997, Russia; olga.griaznova@skoltech.ru (O.Y.G.); yarbel9@mail.ru (I.B.B.); annasogomonyan2012@mail.ru (A.S.S.); 2Center for Photonic Science and Engineering, Skolkovo Institute of Science and Technology, 3 Nobel Str, Moscow 121205, Russia; d.gorin@skoltech.ru; 3Institute for Physics and Engineering in Biomedicine (PhysBio), National Research Nuclear University MEPhI (Moscow Engineering Physics Institute), Moscow 115409, Russia; gvtikhonovskii@mephi.ru (G.V.T.); aapopov@mephi.ru (A.A.P.); nikitin@kapella.gpi.ru (P.I.N.); andrei.kabashin@univ-mrs.fr (A.V.K.); 4Faculty of Physics, M.V. Lomonosov Moscow State University, Moscow 119991, Russia; komlev.as16@physics.msu.ru; 5Prokhorov General Physics Institute of the Russian Academy of Sciences, Moscow 119991, Russia; 6Campus de Luminy—CNRS, LP3, Aix Marseille University, Case 917, 13288 Marseille, France

**Keywords:** pulsed laser ablation in liquids, multimodal imaging, MRI, CT, photothermal therapy, iron-gold nanoparticles, pharmacokinetics

## Abstract

Hybrid multimodal nanoparticles, applicable simultaneously to the noninvasive imaging and therapeutic treatment, are highly demanded for clinical use. Here, Fe-Au core-satellite nanoparticles prepared by the method of pulsed laser ablation in liquids were evaluated as dual magnetic resonance imaging (MRI) and computed tomography (CT) contrast agents and as sensitizers for laser-induced hyperthermia of cancer cells. The biocompatibility of Fe-Au nanoparticles was improved by coating with polyacrylic acid, which provided excellent colloidal stability of nanoparticles with highly negative ζ-potential in water (−38 ± 7 mV) and retained hydrodynamic size (88 ± 20 nm) in a physiological environment. The ferromagnetic iron cores offered great contrast in MRI images with r_2_ = 11.8 ± 0.8 mM^−1^ s^−1^ (at 1 T), while Au satellites showed X-ray attenuation in CT. The intravenous injection of nanoparticles enabled clear tumor border visualization in mice. Plasmonic peak in the Fe-Au hybrids had a tail in the near-infrared region (NIR), allowing them to cause hyperthermia under 808 nm laser exposure. Under NIR irradiation Fe-Au particles provided 24.1 °C/W heating and an IC_50_ value below 32 µg/mL for three different cancer cell lines. Taken together, these results show that laser synthesized Fe-Au core-satellite nanoparticles are excellent theranostic agents with multimodal imaging and photothermal capabilities.

## 1. Introduction

Magnetic nanoparticles (NPs) consisting of iron, gadolinium, manganese, and other metals are widely studied for biomedical applications, e.g., drug delivery, magnetic resonance imaging (MRI), or visualization with magnetic spectral approaches [1,2,3,4]. However, independently of doping metals (Mn, Co, and Ni), most magnetic NPs are iron oxide based. After accumulation in an organism, NPs can degrade in lysosomal conditions in cells, releasing metal ions [5]. It can reduce contrasting properties of NPs and cause adverse toxic effects [6,7]. It was shown that T_1_ MRI contrast agents based on chelates of gadolinium can release Gd^3+^ ions, which subsequently accumulate in brain, muscles, and other tissues, causing apoptosis, the competition of Gd^3+^ and Ca^2+^, and other cytotoxic effects [8]. However, iron-based MRI contrast agents tend to have lower toxicity since there is a plethora of mechanisms of iron metabolism in the body. Moreover, a lysosomal-induced release of Fe^3+^ ions is used for the treatment of iron deficiency anemia [9].

Plasmonic NPs, such as gold, silver, TiN, ZrN, and HfN-based materials have unique optical properties such as increased extinction at resonant frequencies [10]. This property can be used for surface-enhanced Raman spectroscopy (SERS) sensing and visualization [11,12], as well as for local hyperthermia treatment of wound infection or tumors [13,14], which allows avoiding the usage of chemotherapy and, thus, lowers toxicological effects to a body. On the other hand, light absorbance causes an increase in temperature, subsequent nanoparticle volume expansion, and hence the generation of acoustic waves, which is a basis for photoacoustic visualization [15]. Among other plasmonic NPs, the non-reactivity of gold makes it low in toxicity, while a high atomic number renders the X-ray attenuation coefficient high enough for the in vivo visualization of NPs by computed tomography (CT) [16].

The combination of magnetic and plasmonic properties of iron and gold nanoparticles in one core-shell or alloy system renders possible not only multiple visualizations of pathologies by MRI and CT techniques for increasing diagnosis quality but also allows the performance of hyperthermia treatment of cancer cells. Recently, several chemically synthesized Fe-Au composites were investigated for multimodal in vivo imaging [17,18,19]. The chemical preparation of bimetallic nanoparticles can provide a wide variety of different crystal structures and compositions; however, their surface can be contaminated by absorbed reagents and surfactants, which potentially increase nanoparticle toxicity. To solve this problem, the pulsed laser ablation in liquids (PLAL) method was developed [20], which provides nanoparticles with contaminated-free surface and can generate particles with various composition and size. Different PLAL-fabricated particles, including silicon, gold, and TiN nanomaterials, were investigated for biomedical applications and showed low toxicity in vitro and in vivo [21,22,23,24]. A recent synthesis approach of Fe-Au core-satellite hybrid nanoparticles by the PLAL method was proposed by Popov A.A. et al. [25]. It provided unform decorations of nanosized iron cores with smaller gold nanoparticles. This type of Fe-Au nanocomposite assembly is preferred for in vivo applications due to the high biocompatibility of both components—iron oxides are biodegradable in physiological conditions via iron-recycling pathways, while gold nanoparticles are biodegradable in lysosomes during 2–6 months following biomineralization [26]. The resultant uncoated material has strong magnetization (12.6 emu/g) and broad plasmon peaks centered at 550 nm with a long tale in the near-infrared region [25]. These first data provide promising multimodal imaging and photothermal therapy relative to cancer cells using laser-synthesized Fe-Au hybrids, but the potential of this new material has yet to be confirmed in vitro and in vivo.

In this paper, the syntheses of bimetallic Fe-Au core-satellite nanoparticles by the PLAL method and the investigation of their biomedical applications in terms of their MRI, CT contrast properties, and photothermal therapeutic efficiency under irradiation by near-infrared red (NIR) light were reported. Polyacrylic acid coated Fe-Au particles showed low toxicity to cancer cells of different tissues in vitro. Under NIR irradiation, nanoparticles were heated and caused 100% cell death at concentrations of particles higher than 25 µg/mL. Moreover, Fe-Au particles acted as negative T_2_ contrast agents in MRI and positive contrast material for CT imaging in vitro and in vivo. Our results make laser-synthesized Fe-Au core-satellite nanoparticles a perspective agent for multimodal imaging and therapy.

## 2. Materials and Methods

### 2.1. Materials

#### 2.1.1. Chemicals

Sigma-Aldrich (St. Louis, MO, USA): sodium chloride (≥99%), potassium ferrocyanide (≥98.5%), hydrochloric acid (37%), nitric acid (70%), poly(acrylic acid sodium salt) (Mw ~5100) (>86%), 3-(4,5-Dimethyl-2-thiazolyl)-2,5-diphenyl-2H-tetrazoliumbromide (MTT) (≥97.5%), and eosin (99%); Chimmed (Moscow, Russia): dimethyl sulfoxide (DMSO) (99.9%), crystal violet (98%), and formaldehyde (37%); Gibco (Waltham, MA, USA): DMEM medium; Capricorn (Ebsdorfergrund, Germany): fetal bovine serum; PanEko (Moscow, Russia): L-glutamine, penicillin, and streptomycin; Virbac (Carros, France): Zoletil; Bioveta (Komenského, Czech Republic): Rometar.

Targets for laser ablation: Au target (99.99%, GoodFellow, Delson, QC, Canada) and metallic Fe target (99.99%, GoodFellow, Delson, QC, Canada).

#### 2.1.2. Cell Lines

Human tumor cell lines: lung carcinoma A549 (CCL-185™; ATCC), mammary ductal adenocarcinoma BT-474 (HTB-20™; ATCC), and ovarian adenocarcinoma SKOV3-1ip (collection of Laboratory of Molecular Immunology IBCh RAS); mouse mammary cell line EMT6/P (ECACC) and Chinese hamster ovary cell line CHO (Russian Cell Culture Collection) were used for in vitro studies. All cell lines were grown in colorless DMEM medium with 10% fetal bovine serum and 5000 U/mL and 5000 μg/mL of penicillin and streptomycin in culture flasks at 5% CO_2_ and 37 °C. 

### 2.2. Synthesis of Fe-Au Nanoparticles

The synthesis of Fe-Au core-satellite nanoparticles was performed by a two-step method of femtosecond (fs) pulsed laser ablation in liquids. A schematic diagram of the synthesis procedure is shown in Figure 1. At the first step of PLAL, an aqueous solution of Au NPs was synthesized. The target was placed vertically in a glass cuvette filled with 50 mL of 1mM NaCl aqueous solution. The thickness of liquid layer between the target surface and cuvette wall was 4 mm. Radiation from a Yb:KGW laser (1030 nm wavelength, 250 fs pulse duration, 30 µJ pulse energy, 100 kHz repetition rate; TETA 10 model, Avesta, Moscow, Russia) with 3 mm beam diameter was focused by a 100 mm F-theta lens on the surface of the target, through a side wall of the ablation vessel. To avoid ablation from one place and increase synthesis efficiency, the laser beam was moved over a 10 × 30 mm area on the surface of the target, with 4 m/s speed using a galvanometric scanner. The duration of the first step of laser ablation experiment was 40 min. Then, the prepared colloidal solution was centrifuged with 15,000× *g* for 15 min and the nanoparticles in the supernatant were taken for further steps. During the second step of PLAL a metallic Fe target was placed in the chamber filled with Au NP solution. The ablation of Fe target was performed in the same manner as for the first step. The duration of the second step was 15 min.

Non-reacted Au NPs were ejected by a step of magnetic separation. For this, a NdFeB cylindric (diameter—5 cm; length—3 cm) magnet was placed for 5 min next to the bottom of the cuvette with a colloidal solution of Fe-Au NPs. The magnetic fraction of NPs was attracted to the vessel’s bottom, while nonmagnetic NPs remained dispersed in the solution. Then, the upper part of the solution containing free Au NPs and nonmagnetic Fe NPs was discarded, leaving 2 mL of solution at the bottom. The NPs were redispersed by an ultrasonication.

Obtained Fe-Au nanoparticles were stabilized by polyacrylic acid. Polymer solution measuring 1 mL (50 mM) was heated to 80 °C and added to 1 mL of Fe-Au nanoparticle water dispersion with a concentration of 1 g/L. Then, the nanoparticles were incubated for 15 min at 80 °C and washed 3 times with water from unbound polymer via centrifugation at 4000× *g* for 10 min.

### 2.3. Nanoparticle Characterization

Hydrodynamic size and ζ-potential measurements were performed using a Malvern Zetasizer Nano ZS (Malvern Instruments, Malvern, UK). The experiments were performed in distilled water for hydrodynamic size measurement and in 10 mM NaCl for ζ-potential analysis. Number distribution was used for size analysis. Stability measurements were carried out in phosphate-buffered saline (PBS, 100 mM, pH 7.4) at 60 µg/mL concentration.

UV-Vis spectra were measured by an Infinite M1000 PRO (Tecan, Salzburg, Austria) microplate reader.

Surface morphology and the elemental composition of NPs were characterized by a scanning electron-microscope (SEM) MAIA3 (Tescan, Brno, Czech Republic) coupled with an energy dispersive X-ray spectroscope (EDS) X-act (Oxford Instruments, Abingdon, UK). Nanoparticle solution in water was dropped on a silicon wafer and dried under ambient conditions. Nanoparticle images were obtained at 20 kV accelerating voltage in secondary electron (SE) detection mode. The mean size of the Fe-Au core satellites was measured using ImageJ software; 210 nanoparticles were analyzed.

The concentrations of iron and gold in Fe-Au NPs were quantified via inductively coupled plasma mass-spectrometry using a NexION 2000 spectrometer (Perkin Elmer, Waltham, MA, USA). Particles were dissolved in concentrated aqua regia, 3:1 mixture of HCl:HNO_3_, and diluted by water to 30% acid concentration for measurements. ^57^Fe and ^197^Au peaks were used for the analysis.

Magnetic hysteresis loop was measured using a VSM LakeShore 7407 Series magnetometer (Lake Shore Cryotronics, Westerville, OH, USA). The field dependence of magnetization was measured for NP water dispersion sealed in a quartz capillary at room temperature. The measurement was carried out up to 2 T field. Diamagnetic background from the quartz tube and holder were subtracted from measurement results.

### 2.4. Study of Photothermal Properties

To evaluate the photothermal properties of PAA-coated Fe-Au NPs, 1 mL of nanoparticle solution in water was irradiated in a square optical polystyrene cuvette either with 808 nm or 532 nm laser at room temperature. Temperature measurements were performed using a thermal imaging camera FLIR C3 (FLIR Systems, Wilsonville, OR, USA), and the temperature of the hottest spots in the irradiated region were used for analysis. The test on photostability was performed via repetitive cycles of 5 min heating under irradiation with an 808 nm laser (1.23 W) followed by a 5 min cooling period.

Photothermal conversion efficiency (η) of nanoparticles under near infrared irradiation (808 nm, 1.23 W, 10 min duration) was calculated via an approach developed in [27]:$$\eta \left(\%\right)=\frac{hS\left(T_{max}^{NPs}-T^{amb}\right)-Q_{s}}{I\;\left(1-10^{-A^{808}}\right)\;}$$
where h is the heat transfer coefficient, S is the surface area for heat transfer, $T_{max}^{NPs}$ is the maximum temperature reached in particle suspension under irradiation, $T^{amb}$ is the ambient temperature, $Q_{s}\;$is the heat produced by water due to light absorption, I is the incident laser intensity, and $A^{808}$ is the light absorbance of particles at λ = 808 nm. To derive hS, the following relation was used:$$hS=\frac{m_{s}c_{s}}{\tau }$$
where $m_{s\;}$and $c_{s\;}$are mass and specific heat capacity of water, respectively, and τ is the time constant determined as the slope coefficient in cooling time vs. natural logarithm of the driving force ($\theta =\left(T-T^{amb}\right)/$($T_{max}^{NPs}$ − $T^{amb}$)) dependence. The value of $Q_{s}$ was determined by measuring temperature increments in a cuvette filled with distilled water under the same irradiation conditions.

### 2.5. In Vitro Studies

The 3-(4,5-dimethylthiazol-2-yl)-2,5-diphenyltetrazolium bromide (MTT) test was used to determine the cytotoxic effect of Fe-Au@PAA nanoparticles. For this, A549, BT474, SKOV3-1ip, EMT6/P, and CHO cells at a concentration of 6 × 10^4^ cells in 600 μL of colorless DMEM medium were incubated with particles at concentrations of 3.12, 6.25, 12.5, 25, 50, and 100 μg/mL for 1 h at 5% CO_2_ atmosphere and 37 °C. After that, cells were introduced into a 96-well plate at a concentration of 10^4^ cells in 100 μL of medium per well. Cells were incubated for 48 h at 5% CO_2_ and 37 °C. Next, the medium was removed, 100 μL of the MTT solution per well was added, and cells were incubated for 1 h at 37 °C. Then, MTT was removed and 100 µL of DMSO was added to each well. The measurement was carried out on the Infinite M1000 Pro spectrophotometer at a wavelength of 570 nm. Cell viability is shown as a percentage normalized relative to the control cells incubated without particles.

For the measurement of cytotoxicity under near-infrared irradiation, 6 × 10^4^ cells of different lines, A549, EMT6/P, or CHO, were added to 1.5 mL tubes and incubated in 600 μL of colorless DMEM medium with particles at concentrations from 3.12 to 100 mg/L for 1 h at 5% CO_2_ and temperature 37 °C. Next, tubes were covered with foil and samples were irradiated with NIR 808 nm laser (0.76 W) at room temperature while continuously shaking (285 rpm). Then, the irradiated cells were transferred into a 96-well plate at a concentration of 10^4^ cells in 100 μL of medium per well. Cells were incubated for 48 h at 5% CO_2_ atmosphere and 37 °C. Then, cell viability was measured by a MTT test, as described above. Cell viability is shown as a percentage normalized to control cells, incubated without particles and irradiation. The IC_50_ values were calculated using the OriginPro 2015 software (OriginLab, Northampton, MA, USA) with a dose–response function.

For setting clonogenic analysis, the same conditions were used as for the study of cytotoxicity by the MTT test. After the incubation of cells with nanoparticles and their irradiation, the cells were diluted 30 times, and 2 × 10^3^ cells in 1 mL of DMEM medium were added to the each well of a 12-well plate and incubated for 8 days at 5% CO_2_ and 37 °C. Next, the nutrient medium was removed, cells were washed with 600 µL of PBS, pH 7.4. Then, 600 µL of 70% ethanol was added to the wells to fix the cells, following incubation at room temperature for 15 min. Then, 70% ethanol was removed and 600 µL of 95% ethanol was added, repeating incubation conditions. After removal of 95% ethanol, the plates were washed with water. Then, the cells were stained by 600 μL of 1% crystal violet water solution for 30 min at room temperature. Then, the wells of the plates were washed 10 times with water. 

### 2.6. Animals

All animal studies were approved by the Institutional Animal Care and Use Committee (IACUC) of Shemyakin–Ovchinnikov Institute of Bioorganic Chemistry (Moscow, Russia), protocol № 240 from 16 June 2020.

Female BALB/c mice of 18–22 g weight were obtained from Pushchino Animal Facility (Pushchino, Russia) and maintained in a vivarium at the Shemyakin-Ovchinnikov Institute of Bioorganic Chemistry. Before the experiments, mice were anesthetized by an intraperitoneal injection of 150 μL of Zoletil/Rometar solution in a concentrations 90 g/L and 0.16 g/L, respectively.

To perform a tumor model, EMT6/P mice mammary carcinoma cells were inoculated in animals. EMT6/P cells were grown in DMEM supplemented with 10% fetal bovine serum and 2 mM L-glutamine at 37 °C in 5% CO_2_ atmosphere. The cells were harvested from the culture dish and transferred to PBS to obtain the concentration of 10^7^ cells per mL. Then, 100 μL of EMT6/P cell suspension was administered subcutaneously into the flank region of BALB/c mice. Mice bearing tumors of 150–250 mm^3^ volume were used for the experiments.

### 2.7. Pharmacokinetics Study

A magnetic particle quantification (MPQ) method [28] was used to evaluate Fe-Au nanoparticle biodistribution and blood circulation kinetics as described elsewhere [29]. Mice were anesthetized and their tail was placed into the magnetic coil of an MPQ reader and gently fixed with tape. Then, 1 mg of Fe-Au@PAA particles in 100 μL of PBS was injected into the retro-orbital sinus. The concentration of nanoparticles in blood was measured with MPQ in real-time in tail veins and arteries of mice. The 10–90% central part of kinetic curves was fitted with a monoexponential function to calculate the blood circulation time of magnetic NPs.

For the investigation of nanoparticle biodistribution, 2 h after particle administration, mice were euthanized by cervical dislocation, and the major organs (liver, spleen, lungs, kidneys, heart, femur bones, brain, muscle sample, and tumor) were harvested and placed into a measuring coil of a MPQ reader. To increase delivery to the tumor site, magnetic field-assisted delivery was used by placement of a 2 cm × 1 cm × 0.5 cm NdFeB magnet near the tumor. To consider differences in the actual injected dose among different animals, the particle quantities in organs were normalized by the sum of the magnetic signals from all studied organs. Each signal was then divided by the organ mass to obtain the signal per gram of tissue.

### 2.8. Magnetic Resonance Imaging

MRI analysis of mice and phantoms was carried out in an ICON 1T MRI system (Bruker, Billerica, MA, USA) using a mouse whole-body-volume radio frequency coil. For image acquisition, a two-dimensional gradient echo FLASH sequence was used with the following parameters: repetition time/echo time (TR/TE), 1000/5.16 ms; flip angle (FA), 60°; resolution, 200 μm; field of view (FOV), 85 × 30 mm; 1 signal average; 20 slices per scan; slice thickness, 1.5 mm. T_2_ mapping was used for acquiring T_2_ values of NP suspensions.

### 2.9. Computed Tomography

IVIS Spectrum CT (PerkinElmer, Waltham, MA, USA) small animal imaging system was used for computed tomography of phantoms and mice. Images were acquired with 50 kV tube at 1 mA current with 20 milliseconds of exposure time. A total of 720 projections spaced 0.5° apart were acquired, and the CT volume was reconstructed using Living Image software (PerkinElmer Inc., Waltham, MA, USA) with a FOV 8 cm × 8 cm × 2 cm.

### 2.10. Histological Analysis

Samples of the organs were fixed in 4% formaldehyde for 24 h, dehydrated in a series of alcohol solutions of increasing alcoholic concentration, and embedded in paraffin. Paraffin sections measuring 4 μm thick were stained with eosin and Perls Prussian Blue. Microscopy studies were carried out using Leica DM4000 B LED light microscope (Leica Microsystems, Wetzlar, Germany) equipped with a digital camera Leica DFC7000T.

### 2.11. Statistical Analysis

All experiments were performed at least in triplicates. All numerical data are presented as mean ± standard deviation. Significant differences were determined using 2-tailed Welch’s t-test. The statistical difference was considered statistically significant when *p* value was < 0.05 (*), 0.01 (**), and 0.001 (***).

## 3. Results and Discussion

### 3.1. Nanoparticle Synthesis and Characterization

Highly pure Fe_x_O_y_@Au nanoparticles (Fe-Au NPs) were synthesized by a two-step femtosecond pulsed laser ablation in liquid method according to a previously reported procedure [25]. The resulting nanoparticles had core-satellite architecture, where spherical iron oxide NPs formed cores, while Au NPs formed satellites (Appendix A). The mass ratio of Fe:Au was 2.1 ± 0.3, according to analysis by inductively coupled mass spectrometry.

For in vivo applications of nanoparticles, they should possess colloidal stability in high ionic strength liquids. As-prepared Fe-Au nanoparticles had mean hydrodynamic size of (73 ± 40) nm in water, while in physiologically relevant environment of phosphate buffered saline (PBS, 100 mM, pH 7.4), a rapid aggregation to sub-micron range was observed with hydrodynamic size of (673 ± 200) nm (Figure 2a). To improve the poor colloidal stability of Fe-Au, NP surface was coated with a biocompatible organic polymer—polyacrylic acid (PAA). PAA is a polymer, which bears all qualities suited for biomedical applications, such as biocompatibility, nontoxic nature, and biodegradability [30]. The obtained PAA-coated Fe-Au core-satellite nanocomposites (Fe-Au@PAA) possessed slightly higher diameter in water (88 ± 40) nm, indicating the appearance of an additional polyelectrolyte kinetic layer. The PAA coating effectively prevented nanoparticle aggregation in PBS for at least 1 day, which was confirmed by the retainment of the mean particle size at the same value: (88 ± 20) nm (Figure 2b).

The formation of the polyelectrolyte layer of PAA on the surface of Fe-Au nanoparticles was proved by changes in ζ-potential from (+10 ± 5) mV before to (–38 ± 7) mV after the coating procedure (Figure 2c). Low positive ζ-potential of bare NPs indicated weak interparticle repulsion. According to theory [31], nanoparticles, which have ζ-potential less than 5 mV by modulus, undergo aggregation, while |ζ| > 30 mV provides a good colloidal stability of NP solution. Fe-Au@PAA NPs demonstrated a significantly higher ζ-potential magnitude with the negative charge, which is a distinct indication of PAA carboxylate anions presented on the particle’s surface. A 1 mV shift in ζ-potential peak value was observed after 25 days of coated particles incubation at 4 °C (Appendix A), thus showing long-term stability of the organic layer during storage. Considering high chemical stability of both Au nanoparticles and oxidized iron in water, Fe-Au@PAA NPs could provide a particularly long half-life in aqueous colloid form, which is highly relevant for biomedical implementation.

The preservation of core-satellite structure after the coating was confirmed by SEM imaging coupled with EDS mapping, demonstrating the sparse distribution of gold around spherical iron-based cores (Figure 2d,e). The mean physical diameter of cores in the coated nanoparticles was determined to be (74 ± 20 nm) (Figure 2f). The ratio of hydrodynamic and physical sizes indicated that Fe-Au@PAA NPs rather exist in solutions in a single form than in clusters.

Core-satellite nanoparticles possesses both magnetic and plasmonic properties. Magnetic hysteresis loop at 300 K temperature showed ferromagnetic behavior with coercivity ≈ 41 Oe (Figure 2g). Magnetization was measured for 1 mL of water dispersion of Fe-Au NPs with 1 mg/mL concentration. The volume magnetization saturation of Fe-Au NPs was M_S_ ≈ 0.022 emu/cc or 22 emu/g. This value is lower than for bare laser-synthesized iron oxide nanoparticles with M_S_ = 44.7 emu/g [25] for ultrasmall iron nanoparticles Fe-NPs (M_S_ = 117–163 emu/g) and other Fe_3_O_4_-Au (M_S_ = 86 emu/g) hybrids [32,33]. It should be noted that artifacts in magnetization curves at |H| > 0.5 kOe may stem from the aggregation of nanoparticles at high magnetic fields, which is challenging to avoid in liquids.

Plasmonic gold NPs contributed to the extinction spectra of iron oxides by appearance of absorbance peak in green region of the spectra with a broad tail in the infra-red part (Figure 2h), which is in an optical transparency window of biological tissue [25]. After the coating of Fe-Au NPs by PAA, the plasmon peak was shifted from 542 nm to 536 nm with decrease in its intensity, probably due to a slight release of loosely bound plasmon Au nanoparticles during the coating procedure.

### 3.2. Photothermal Properties of Fe-Au@PAA Nanoparticles

Nanoparticles demonstrating strong light absorption in the visible and near-infrared regions and then converting it into heat are promising candidates for photothermal therapy. Thus, owing to the strong extinction of Fe-Au at the plasmon resonance band, the photothermal properties of NPs under irradiation by a green light from a 532 nm laser source were evaluated firstly. Green lasers have been extensively used in surgery for photocoagulation [34] and tested for cancer therapy with nanoparticles [35]. Considering relatively low intensities of 532 nm lasers used in clinics (100–750 mW) [36] and high absorption of visible light by tissues, Fe-Au@PAA NPs were exposed to irradiation with 100 mW laser power. Under the irradiation, the steady-state temperatures increased from 26.5 to 31 °C with the increase in particle concentration from 25 to 400 µg/mL (Appendix A). This result suggests the weak opportunities of these particles to cause even mild hyperthermia under such irradiation conditions.

Nevertheless, the intense extinction in the near-infrared region associated with the light absorption by iron cores and clustered Au satellites makes Fe-Au NPs attractive for photothermal therapy with an 808 nm laser. This wavelength belongs to the first biological transparency window (650–950 nm), where low absorption and scattering of light by tissues provide highest light penetration depth [37]. Indeed, the aqueous suspensions of Fe-Au@PAA NPs under irradiation with a continuous 808 nm laser demonstrated excellent thermal response at concentrations 10–400 µg/mL (Figure 3a,b). For example, the heating of particles at 50 µg/mL from temperature of 37 °C to the conventional upper temperature limit of mild hyperthermia (42 °C) was reached just after 2.5 min of irradiation. The saturation in heating efficiency clearly appeared after 100 µg/mL NP concentration and reached 39 °C temperature increase at concentrations of 400 µg/mL. A linear dependence of heating efficiency on laser power was observed (Figure 3c). The high value of the slope coefficient (24.1 °C/W) suggests that particles can provide high photothermal efficiency under even low NIR light intensities that should be safe for healthy tissues.

Fe-Au NP suspension demonstrated a decent photothermal stability under on/off irradiation cycles, although particle sedimentation during the heating caused a slight decrease in peak temperature at the fifth cycle. The time constant of suspension cooling τ = 318 s was used for calculation of photothermal conversion efficiency coefficient (η) (Figure 3e,f). Fe-Au@PAA particles show η = 38%; thus, their efficiency exceeds those reported for gold nanoparticles (10.2%) [38], gold nanoshells (13%) [34], other iron-oxide nanoparticles with modified surface (13.1–35.7%) [39], and is comparable to another Fe-Au core-shell nanocomposites (42.6%) [17] evaluated under NIR irradiation.

### 3.3. In Vitro Studies of Fe-Au@PAA Biocompatibility and Toxicity after the Photothermal Treatment

The MTT assay [40] was used to evaluate the cytotoxic effects of Fe-Au@PAA nanoparticles on cell viability. This colorimetric test is based on the cellular reduction of tetrazolium dye (3-(4,5-dimethylthiazol-2-yl)-2,5-diphenyl-tetrazolium bromide), where living cells recover yellow colored tetrazolium dye to insoluble purple colored formazan in response to dehydrogenase activity. The MTT test measures cell viability in terms of reductive activity since the enzymatic conversion of tetrazolium occurs only in mitochondria and other organelles of living cells [41].

Five cell lines of different tissues and organisms were used in toxicity study: human (A549, SKOV3-1ip, BT-474), mouse (EMT6/P), and Chinese hamster (CHO) cell lines. To evaluate cytotoxic effects of Fe-Au@PAA, NPs were incubated with cells for 48 h as described in the Materials and Methods section.

The cytotoxicity of Fe-Au@PAA NPs at concentrations 3.12–50 μg/mL in BT-474 and SKOV3-1ip cells within tested time was negligible. A slight decline in cell viability was observed for 100 μg/mL concentration of nanoparticles with cell viability values being 67 ± 9% and 82 ± 3% for BT-474 and SKOV3-1ip cell lines, respectively (Figure 4a).

Nanoparticles possessed slight dose-dependent cytotoxic effects on A549 and CHO cells (Figure 1b–d green columns). However, cell viability for the 50 μg/mL dose was 66 ± 3% (CHO), 79 ± 5% (A549); for the 100 μg/mL dose, it was 21.6 ± 0.6% (CHO) and 73 ± 3% (A549). Note that, even at high concentrations of nanoparticles, cell viability was above 70%, which is consistent with the results obtained by other authors [17,42]. It confirms the biocompatibility of obtained Fe-Au@PAA nanoparticles for the studied cells.

The study of the effectiveness of photothermal treatment via Fe-Au@PAA nanoparticles was performed on three cell lines: A549, EMT6/P, and CHO (Figure 4b–d). To evaluate effects from laser-induced hyperthermia, cells were incubated with NPs at different concentrations for 1 h, following sample irradiation by 808 nm laser with a 0.76 W power for 7 min. These conditions were determined experimentally to not cause photothermic damage to cell lines without NPs (Figure 4b–d, see red columns at “0” NP dose).

Almost 100% of cell death was observed at concentrations 50 and 100 μg/mL for all cell lines. IC_50_ of Fe-Au@PAA NPs with irradiation for A549 cells was (32 ± 1) μg/mL; EMT6/P— (29 ± 3) μg/mL; and CHO— (21 ± 1) μg/mL, while the IC_50_ of NPs without exposure by laser could not be determined. Fe-Au@PAA NPs apparent had the most cytotoxic effects on the CHO cell line with and without laser exposure. The higher toxicity induced by photothermal treatment was caused by CHO higher sensitivity to heating, as was shown by other authors [43].

The MTT test reflects cellular activity, while clonogenic analysis is a method for determining the survival and proliferation of cells for a long period after their exposure to various agents [44,45]. Thus, a clonogenic assay was used to assess the long-term cytotoxic effects as it is a more sensitive test than MTT. The clonogenic test was carried out under the same experimental conditions. Cell lines: EMT6/P and CHO were incubated with Fe-Au@PAA NPs following irradiation by 808 nm laser. After dilution and incubation of cells for 8 days, the growing cell colonies were stained with crystal violet, and the results were evaluated visually. Cells without NP exposure and light irradiation were used as controls (Figure 5). The considerable inhibition of cell proliferation was observed at NP doses higher 12.5 µg/mL after the laser induced hyperthermia for both cell lines. At two-fold higher doses, no forming colonies were observed. There was no decrease in proliferative activity at lower concentrations of nanoparticles—3.12 and 6.25 μg/mL. Thus, the clonogenic assay showed that even smaller concentrations of Fe-Au nanoparticles can damage cancer cells under heating in comparison to the MTT test. In addition, while the MTT test of cell viability showed moderate cytotoxic effects of Fe-Au@PAA NPs to CHO cells with 21.6% enzyme activity at 100 μg/mL particle dose, clonogenic analysis did not confirm such dramatic toxicity (Figure 5). Thus, nanoparticles at high doses have biocompatibility in terms of cell survival but can affect enzyme activities of cells.

Despite the demonstrated promise of the photothermal properties and low in vitro toxicity of Fe-Au core-satellite nanoparticles, in vivo toxicity issues were left behind and warrant further studies. Natural and synthetic inorganic nanoparticles potentially can cause toxic effects for the organism. One of the common mechanisms is the formation of reactive oxygen species, which lead to increases in the level of lipid peroxidation, cause oxidative DNA damage, and result in carcinogenesis. However, the composition and physicochemical properties of NPs greatly alter toxicity caused by NP exposure. A large number of studies reported that iron oxide nanoparticles regardless of their physicochemical properties may cause cytotoxicity only at concentrations of 100 g/mL in specific organ sites, which is difficult to achieve in the organism [46]. In addition, many clinical studies of iron oxide-based NPs showed that such particles can be categorized as biocompatible. For example, different iron oxides are used for iron-deficient anemia treatment in humans with doses up to 7 mg/kg of body weight without notable toxicity [47]. The toxicity of gold nanoparticles highly depends on size, shape, and chemical groups on their surface. However, many anticancer drugs or diagnostic tools based on gold or gold composite NPs are being tested in clinical trials with several gold NP-based technologies recently approved [48]. In 2019, gold nanoshells were successfully investigated for the photothermal treatment of prostate cancer in humans in a 36 mg/kg dose with rare advert events reported during 3 month after the treatment [49]. As such, we believe that Fe-Au composites can also show low toxicity in vivo in doses required for photothermal therapy and imaging.

### 3.4. Pharmacokinetics of Fe-Au@PAA Nanoparticles

Due to the ferromagnetic nature of Fe-Au nanoparticles, it was possible to non-invasively measure their concentration in vivo via MPQ (magnetic particle quantification) method [28]. This spectral approach allows the quantitative detection of superparamagnetic and ferromagnetic nanoparticles without the interference of diamagnetic or paramagnetic materials that are naturally presented in the organism [29].

To measure kinetics of Fe-Au@PAA NP circulation in blood, mice were anesthetized, animal’s tail was placed into the magnetic coil, and nanoparticles were administered into the retro-orbital sinus at a 40 mg/kg dose. After that, a rapid increase in magnetic signal was observed, indicating the entry of NPs to the bloodstream, following by the monoexponential decrease due to the nanoparticle’s clearance (Figure 6a). After 20 min of circulation Fe-Au@PAA nanoparticles were almost completely eliminated from the bloodstream, with circulation half-life time t_1/2_ = (4.4 ± 0.6) min. It is well known that polyacrylic acid coated NPs have small blood circulation time since this polymer did not protect nanoparticles from the adsorption of opsonins in serum [50].

To increase NP accumulation in tumor, magnetic field-assisted delivery was implemented by placement of 2 cm × 1 cm × 0.5 cm NdFeB magnet near the tumor. Biodistribution of NPs 30 min after injection was studied by analyzing magnetic signals in main organs, such as liver, spleen, lungs, femur bones, kidneys, heart, muscles, brain, and tumor (Figure 6b). Major part of Fe-Au@PAA NPs accumulated in the liver (77 ± 4% ID/g) and spleen (39 ± 9% ID/g), which are the main organs of mononuclear phagocyte system (MPS). The histological slices of liver and spleen stained by Perls Prussian blue and eosin indicated uptakes of Fe-Au@PAA NPs by Kupffer cells of liver and by the cells in the marginal zone of spleen (Appendix A). Fe-Au@PAA NP concentration in lungs was 30 ± 10% ID/g. The possible reason is increased particle hydrodynamic size by the formation of protein corona and the partial aggregation of nanoparticles following retention in capillaries. This behavior was supported by observing the aggregates of nanoparticles in histological slices of lung tissue (Appendix A). Fe-Au@PAA NPs were also accumulated in femur bones possibly due to the uptake by bone-marrow derived macrophages or osteoclasts (8 ± 2% ID/g). Less than 2% ID/g concentration of Fe-Au@PAA nanoparticles were observed in kidneys, heart, muscles, and brain.

The concentration of nanoparticles in tumor reached 17 ± 5% ID/g, which is 17-fold higher than in muscles due to magnetic field-assisted delivery and enhanced permeability and retention effects [51]. Tumor vasculature is leaky and possesses larger endothelial gaps and pores than the capillaries of healthy organs. Histology slices of tumor showed that a major part of nanoparticles accumulated at the borders of tumor. Recently, it was shown that the binding of NPs to the extracellular matrix can prevent their penetration deep into the tissue [52]. Nevertheless, most NPs, which were not trapped by cells of MPS, accumulated in tumor sites, providing a contrast in pathology relative to the surrounding tissues.

### 3.5. Magnetic Resonance Imaging

The hybrid Fe-Au@PAA nanoparticles possess all featured properties of theranostic agents, such as biocompatibility, photothermic therapeutic effect for cancer treatment, and dual-modal imaging for diagnostics. To evaluate Fe-Au@PAA NPs as contrast agents for magnetic resonance imaging, T_2_-wheighted images of different concentrations (NP: 1–1000 mg/L; or Fe: 0.009–9 mM) of NPs in water were measured at 1T MR scanner. Fe-Au@PAA NPs decreased signals on T_2_-weighted images and exhibited concentration-dependent negative contrast enhancement effect (Figure 7a). The inverse proton 1/T_2_ relaxation time had non-linear dependence on the concentration of Fe-Au@PAA NPs at studied concentrations. However, the curve could be fitted at lower concentrations with a linear function shown as a black line through the data with a correlation coefficient above 0.98 (Figure 7b). The relaxation rate (r_2_) was determined to be (309 ± 20) s^−1^/(g/L) or (11.8 ± 0.8) s^−1^/mM of iron atoms. This value is lower than those reported in the literature for different structures of Fe_3_O_4_@Au nanoparticles [53,54] due to low crystallinity of laser synthesized Fe_x_O_y_ core of Fe-Au NPs.

For testing in vivo MR contrast properties, 1 mg of core-satellite Fe-Au@PAA NPs was injected into retro-orbital sinus of mice (Figure 7c), and magnetic guided-delivery to the tumor was performed. MR images were made 30 min after the injection to ensure full NP clearance from the blood. Before the injection of nanoparticles, there was no difference in signals of tumor and muscles, while after the injection NPs showed negative contrast in a border region of the tumor as well as in volume. However, nanoparticles were mostly accumulated in the liver and spleen, which is in agreement with the biodistribution analysis by the MPQ method. Nevertheless, the amount of NPs trapped in the tumor was enough for pathology visualization, which is beneficial for MRI-guided surgery or external therapy.

### 3.6. Computed Tomography

Gold nanoparticles are widely used as contrast agents for computed tomography since Au has high atomic number (Z = 79) and, thus, strong X-ray attenuation properties [55]. Due to the presence of Au NPs in a satellite shell, which provided ~32% of the particle mass, Fe-Au NPs should enhance X-ray attenuation with positive contrast at computed tomography. To illustrate the capability of Fe-Au@PAA NPs to enhance CT signals, firstly, the phantom images of aqueous solutions of NPs of different concentrations (0.5–20 g/L) were obtained (Figure 8a). The X-rays were generated with a tube at 50 kV voltage and 1 mA current. Results showed that Fe-Au@PAA NPs at concentrations lower than 1 mg/mL (0.026 mM of Au) had no difference in X-ray enhancement compared to distilled water. X-ray attenuation linearly depended on NP concentration with 0.97 Pearson’s correlation coefficient (Figure 8b). The slope value was (57.7 ± 0.4) HU/(g/L) or (3.75 ± 0.02) HU/mM if recalculated to the molar concentration of Au atoms. The slope value of Fe-Au@PAA NPs was higher than that of Iohexol 2.7 HU/mM, a CT contrast agent commonly used in clinics [56].

To test Fe-Au@PAA as CT contrast agents, the in vivo X-ray attenuation properties of nanoparticles were measured before and after intravenous and intratumoral injections. After intravenous injections of 1 mg of Fe-Au@PAA NPs, no significant effects of contrast were achieved (Figure 8c). According to the literature data, sufficient signal enhancement in vivo can be provided after intravenous administrations only of 5–7 mg of pure gold NPs, which is approximately 15-times higher than the Au dose used in this study [57,58]. Intravenous injection in a rat tail of Fe_3_O_4_@Au composite nanoparticles in doses 28-times higher than that used in this work led to the contrast enhancement of the aorta and whole liver [59]. After an intratumoral injection of NPs in 40 µg dose, remarkably improved contrast enhancement was observed, which rendered it possible to visualize boundaries between tumor and normal adjacent tissues on axial projections.

Fe-Au@PAA NPs is a promising candidate for dual MRI and CT imaging. However, for further improvements of contrasting properties of both modalities, one needs to increase the accumulation of NPs at the targeted region. For this, different target proteins for specific tumor types could be anchored to Fe-Au@PAA NPs surface, e.g., antibody, affibody, or darpins [60]. Another approach is to modulate the accumulation of NPs by the prolongation of their blood circulation time by means of surface modification with stealth polymers or cell membrane coating [61,62] or by the blockade of mononuclear phagocyte system cells [63]. The implementation of both strategies at once might greatly increase the efficiency of nanoparticle trapping at a tumor site.

## 4. Conclusions

This study presents the first assessment of biocompatibility, pharmacokinetics, and the application of laser-synthesized Fe-Au NPs for MRI/CT imaging and photothermal treatment. NPs were synthesized via laser ablation in liquids, which enabled the obtainment of materials with chemically pure surfaces. Coating by polyacrylic acid strongly improved the colloidal stability of Fe-Au NPs in a biologically relevant fluid, making them suitable for biomedical applications. Plasmon peaks in the visible region with a long near-infrared tail allowed heating Fe-Au core-satellites under 808 nm laser exposure. At 25 µg/mL concentration, Fe-Au@PAA NPs showed low cytotoxic effects in vitro, while under laser irradiation, they induced 100% cell death. Due to hybrid core-satellite architecture of Fe-Au@PAA, NPs had contrasting properties both in MRI and CT imaging modalities. In vivo injection of NPs allowed the visualization of tumor boundaries in T_2_-weighted MR images and CT scans. Our results provided promise for the development of novel phototherapy and imaging agents, profiting from the superior properties of laser-synthesized bimetallic nanoparticles with core-satellite structure.

## Figures and Tables

**Figure 1 pharmaceutics-14-00994-f001:**
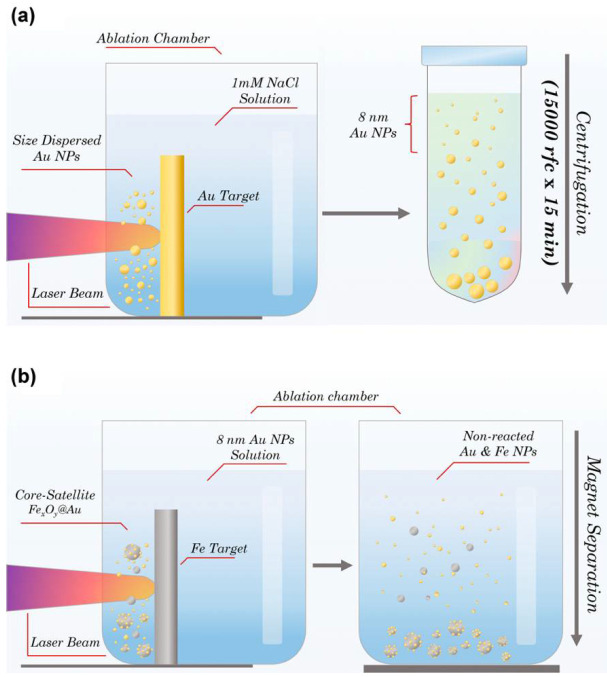
Schematic representation of two-step PLAL synthesis of core-satellite Au-Fe NPs. (**a**) Laser ablation of gold target with further separation step of 8 nm Au NPs fraction via centrifugation. (**b**) Laser ablation of iron target in presence of Au NPs with further separation step of magnetic Fe-Au NPs fraction via external magnetic field.

**Figure 2 pharmaceutics-14-00994-f002:**
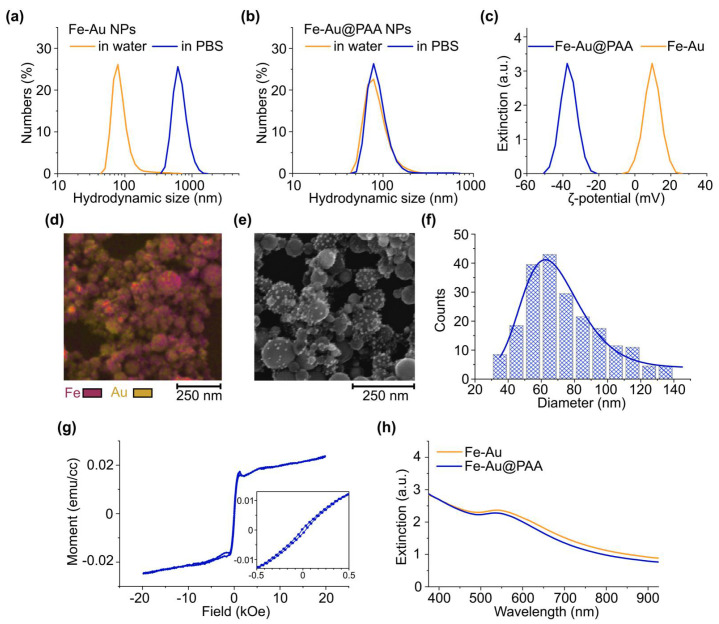
Fe-Au NP characterization before and after stabilization with a polymer coating. (**a**,**b**) Hydrodynamic size distribution of Fe-Au (**a**) or Fe-Au@PAA (**b**) NPs in water and PBS. (**c**) ζ-potential distributions of Fe-Au nanoparticles before and after the coating with PAA. (**d**) EDS mapping of Au and Fe abundance in nanoparticles. Scale bar—250 nm. (**e**) SEM image of Fe-Au@PAA NPs. Scale bar—250 nm. (**f**) Distribution of Fe-Au@PAA NP physical diameter. (**g**) Magnetization curves of Fe-Au NPs measured at 300 K. The inset is an enlarged view of the low-field region. (**h**) UV-Vis spectra of Fe-Au NPs before and after the PAA coating.

**Figure 3 pharmaceutics-14-00994-f003:**
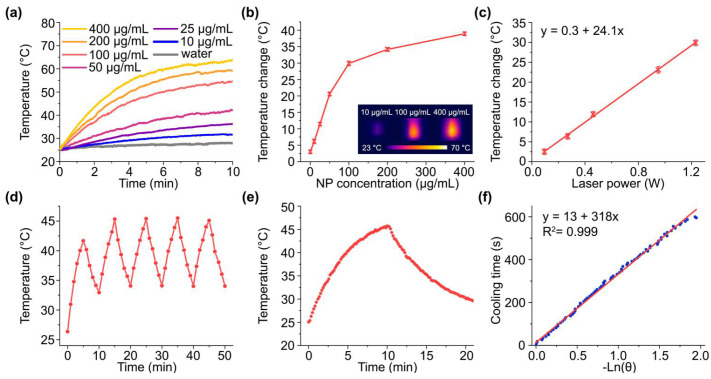
Photothermal properties of Fe-Au@PAA NPs under NIR-irradiation with 808 nm laser. (**a**) Temperature changes for aqueous solutions of NPs under irradiation at 1.23 W laser power. (**b**) Dependence of maximal temperature change on particle concentration. Insert shows thermal images of several particle samples. (**c**) Dependence of temperature change on laser power. (**d**) Photothermal stability cycling test with 5 min heating and 5 min cooling steps. © Kinetics of the temperature changes illustrating 1 cycle of 10 min heating and 10 min cooling. (**f**) The dependence of cooling time on negative natural logarithm of driving force. Solution with 100 µg/mL (**c**) or 50 µg/mL (**d**–**f**) particle concentration was used for the studies.

**Figure 4 pharmaceutics-14-00994-f004:**
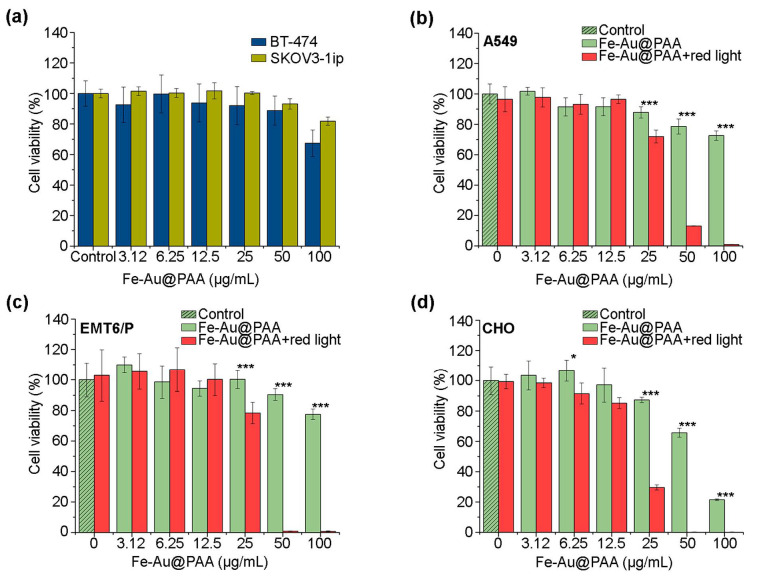
Analysis of the Fe-Au@PAA nanoparticle cytotoxicity using the MTT test. (**a**) Study of the toxic effect of particles upon exposure to cells BT-474 and SKOV3-1ip. (**b**–**d**) Investigation of the photothermic properties of particles. Cells were incubated with Fe-Au@PAA NPs at various concentrations and irradiated for 7 min with an 808 nm laser at 0.76 W power. Cell viability is shown as a percentage normalized to control cells incubated without particles and irradiation. The statistical differences were considered significant when the *p* value was < 0.05 (*), 0.001 (***), Welch’s *t*-test.

**Figure 5 pharmaceutics-14-00994-f005:**
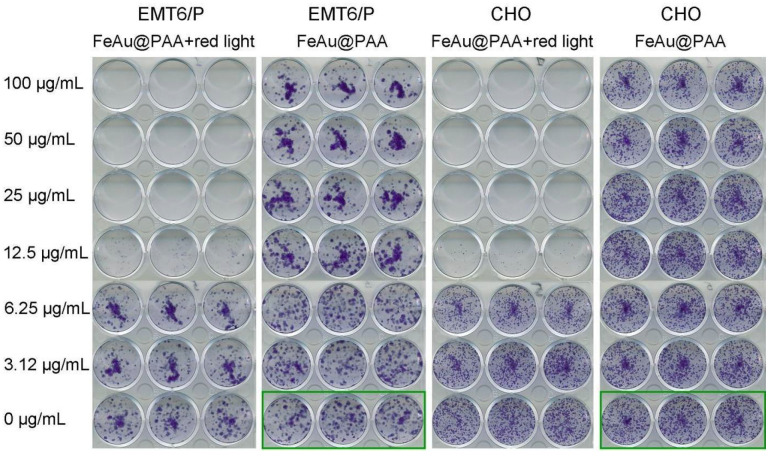
Colony formation assay of EMT6/P and CHO cells treated with Fe-Au@PAA NPs for 8 days with or without 808 nm laser irradiation. Control samples not treated with nanoparticles and irradiation are shown in green frames.

**Figure 6 pharmaceutics-14-00994-f006:**
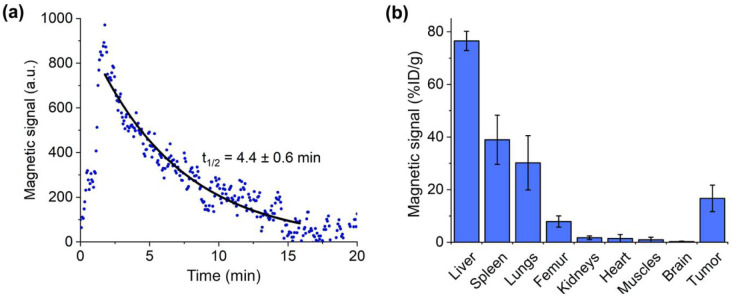
Pharmacokinetics of Fe-Au@PAA NPs. Blood circulation (**a**) and biodistribution (**b**) of nanoparticles.

**Figure 7 pharmaceutics-14-00994-f007:**
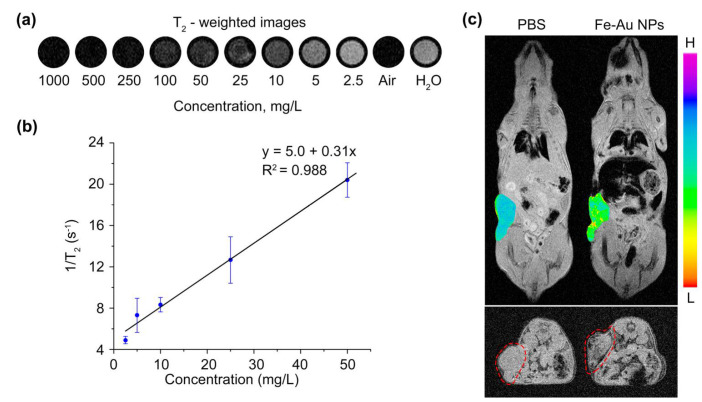
Application of Fe-Au@PAA NPs as MRI contrast agents. (**a**) T_2_-weighted images of NP solutions at 1–1000 mg/L concentrations, distilled water, and air. (**b**) Inverse of the proton relaxation time T_2_ as a function of Fe-Au@PAA NP concentration. (**c**) MR images of EMT6/P tumor bearing mouse after intravenous injection of PBS or Fe-Au@PAA NPs. In the coronal projections (top), a color image of tumor was combined with grey-level image, and scale represents signal intensity. In the axial projections (down), the tumor is bordered by a red dashed line.

**Figure 8 pharmaceutics-14-00994-f008:**
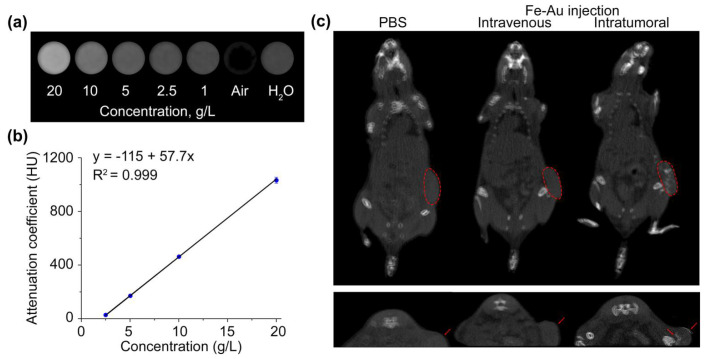
Application of Fe-Au@PAA NPs as CT contrast agents. (**a**) CT images of NP solutions at 1–20 g/L concentrations, distilled water, and air. (**b**) X-ray attenuation coefficient as a function of Fe-Au@PAA NP concentration. (**c**) CT images of EMT6/P tumor bearing mice before and after intravenous or intratumoral injections of Fe-Au@PAA NPs. The tumor boundary is indicated by a red dashed line in coronal projections (top) or with red arrows at axial projections (down).

## Data Availability

All the data are presented within this article and its Appendix A. The raw datasets are available on request from the corresponding authors.

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
