# Peer review of "Laser Synthesized Core-Satellite Fe-Au Nanoparticles for Multimodal In Vivo Imaging and In Vitro Photothermal Therapy"

_pharmaceutics, 2022, doi:10.3390/pharmaceutics14050994_

Round 1

Reviewer 1 Report

In this paper, authors tried to synthesized bimetallic Fe-Au core-satellite nanoparticles by PLAL method and investigated their biomedical applications in terms of their MRI, CT contrast properties and photothermal therapeutic efficiency under illumination of near-infrared red (NIR) light. The paper is well prepared and results are iterating. I recommend a minor revision.

-I have some concerns about toxicity issues of inorganic NPs in human body. Please explain

-what is the nature of negative zeta potential? Where does the negative charge of nanoparticles come from?

-I suggest to added some new interesting paper in introduction: https://doi.org/10.1021/acsami.1c23676, https://doi.org/10.1021/acs.nanolett.0c01757, https://doi.org/10.1016/j.apcatb.2021.120840, https://doi.org/10.1088/1361-6528/abf878, https://doi.org/10.1007/s00339-021-04917-8

-please speak about characteristic properties of prepared NPs (size, morphology, zeta, …) in abstract

-there are some English errors in the text, it needed to check again whole manuscript

-some abbreviation used without explanation please add

Author Response

Dear Reviewer #1,

First of all, we would like to sincerely thank you for evaluation of our manuscript and for the valuable comments. Please find below the answers to the questions mentioned in your review (the Reviewer’s remarks are presented in bold, and our answers follow in plain text, quotations from the manuscript are shown in italic).

-I have some concerns about toxicity issues of inorganic NPs in human body. Please explain

We should draw the Reviewer attention, that some toxicological issues (like in vitro toxicity of Fe-Au particles) were studied and discussed in the text. In the revised version we increased discussion of the toxicity studies and described some clinical studies of iron oxide and gold nanoparticles.

-what is the nature of negative zeta potential? Where does the negative charge of nanoparticles come from?

Fe-Au core-satellite nanoparticles obtained negative zeta potential after coating with polyacrylic acid due to presence of large number of carboxylic groups in the polymer structure, which is mentioned in the manuscript:

“Fe-Au@PAA NPs demonstrated a significantly higher ζ-potential magnitude with the negative charge, which is a distinct indication of PAA carboxylate anions presented on the particle surface.”

-I suggest to added some new interesting paper in introduction:

https://doi.org/10.1021/acsami.1c23676

https://doi.org/10.1021/acs.nanolett.0c01757

https://doi.org/10.1016/j.apcatb.2021.120840

https://doi.org/10.1088/1361-6528/abf878

https://doi.org/10.1007/s00339-021-04917-8

We included several relevant papers in the introduction.

-please speak about characteristic properties of prepared NPs (size, morphology, zeta, …) in abstract

Corrected as follows:

Biocompatibility of Fe-Au nanoparticles was improved by coating with polyacrylic acid, which provided excellent colloidal stability with highly negative ζ potential in water (-38 ± 7 mV) and hydrodynamic size (88 nm) in a physiological environment.”

-there are some English errors in the text, it needed to check again whole manuscript

Corrected.

-some abbreviation used without explanation please add

Corrected.

Reviewer 2 Report

In their report, Deyev and co-workers report the development of multi-modal theranostic nanoparticles enabling in vivo imaging and photothermal therapy. These nanoparticles were used to visualize adenocarcinoma and by the virtue of their plasmonic peak in the NIR region, they could be activated using the 808 nm laser for photothermal therapy. The authors systematically evaluate the in vitro properties using MTT assays and colony-formation assays to find good toxicity. Interestingly the PK profile of these nanoparticles showed a very high liver-to-background uptake in comparison to the tumor-to-background. However, as a first-in-class system of Fe-Au- NPs, this report lays a nice platform for further developments in this area. The reviewer recommends its publication in its current form.

Author Response

We would like to thank Reviewer #2 for the high appreciation of our work and the time he spent on the review.

Reviewer 3 Report

The research work presented in the manuscript pharmaceutics-1685755, entitled “Laser synthesized core-satellite Fe-Au nanoparticles for multimodal in vivo imaging and photothermal therapy" by the group of authors, in the Section: Nanomedicine and Nanotechnology, Special issue: "On the road to precision medicine: magnetic systems for tissue regeneration, drug delivery, imaging, and theranostics" is very well written and justified through suitable evaluation parameters and references. This manuscript contains sufficient novelty to be accepted for publication, but still minor modifications and suggestions are recommended to improve the quality.

The abstract provides an important overview of this research work (background, methods and results) as well as meaningful conclusion.

Introduction gives description of background of the art, as well as goals and expected results.

Materials and Methods part provides necessary data about different applied methods for synthesis of Fe-Au core-satellite nanoparticles, and characterizations (hydrodynamic size and ζ-potential measurements, UV-Vis, scanning electron microscopy coupled with an energy dispersive X-ray spectroscopy, inductively coupled plasma mass spectrometry, magnetic and photothermal properties, magnetic resonance imaging, computed tomography, in vitro studies of Fe-Au@PAA biocompatibility and toxicity after the photothermal treatment, pharmacokinetics study, histological analysis). It is needed to add the purity of used chemicals. The preparation methods of Fe-Au core-satellite nanoparticles were described in sufficient details. All methods were performed by quality equipment.

In the part Results and Discussion all experimental results are in details planed and presented at 8 figures and additional 4 figures in the supplementary material. Authors analyzed and interpreted obtained results according to previous investigations by comparison with cited relevant literature. This research study is a good quality and future research directions were also highlighted.

The conclusions are very good as per good idea presentation and the scientific contribution is visible and applicable.

It is needed to correct the mistake in sections numbers: "4. Results and Discussion" instead of "2. Results and Discussion" and "4. Conclusions" instead of "2. Conclusions".

Also, it is needed to provide the full names of acronyms on first appearance in the abstract, manuscript and in figure captions (e.g. MRI, CT, MRI/CT, NIR, NdFeB...).

It is needed to avoid 1st person plural and rewrite all sentences in 3rd person plural and passive voice (abstract, Introduction, ).

The authors cited 52 papers in manuscript with relevant 10 autocitations.

I recommend the acceptance of the manuscript in the journal Pharmaceutics with minor modifications on above mentioned suggestions and comments.

Best regards,

Reviewer

Author Response

First of all, we would like to sincerely thank you for evaluation of our manuscript and for the valuable comments. Please find below the answers to the questions mentioned in your review (the Reviewer’s remarks are presented in bold, and our answers follow in plain text).

It is needed to add the purity of used chemicals.

Corrected.

It is needed to correct the mistake in sections numbers: "4. Results and Discussion" instead of "2. Results and Discussion" and "4. Conclusions" instead of "2. Conclusions".

Corrected.

Also, it is needed to provide the full names of acronyms on first appearance in the abstract, manuscript and in figure captions (e.g. MRI, CT, MRI/CT, NIR, NdFeB...).

Corrected.

It is needed to avoid 1st person plural and rewrite all sentences in 3rd person plural and passive voice (abstract, Introduction, ).

Corrected.

Reviewer 4 Report

The manuscript by Griaznova et al describes the synthesis and physicochemical characterization of bimetallic core-shell Fe-Au polyacrylic stabilized on hybrid nano-systems for multimodal imaging (MRI and CT) and photothermal therapy in cancer cells.

The synthesis of the core-shell nanostructures was taken place by pulsed laser ablation in liquids (PLAL). This technique represents a very clean and straightforward methodology for affording these kinds of materials. Physico-chemical characterization is robustly built by DLS, Z-Pot, and mass spectrometry besides SEM and EDS mapping.

Magnetic and plasmonic properties were also analyzed, by magnetization curves and UV-spectra respectively, and biological evaluation in vitro with several cell lines was carried out to afford information about the toxicity and biocompatibility after the photothermal treatment by MTT and the long-term effect of the clonogenic assay.

Pharmacokinetics was analyzed by magnetic particle quantification (MPQ) showing high accumulation into solid tumors when a magnetic field is applied in the zone.

Magnetic resonance imaging (MRI) and CT showed contrasting properties.

Briefly, the work is well resolved and interesting for the nanomedicine field and deserves to be published in Pharmaceutics.

Author Response

We would like to thank Reviewer #4 for the high appreciation of our work and the time he spent on the review.

Reviewer 5 Report

This work presents the synthesis and in vivo imaging/therapy testing of Fe-Au nanoparticles. The paper was well-written, the figures were clear and the experiments were well-design. I highly recommended the paper for publication on Pharmaceutics journal. My minor comments are as below:

  1. The title includes the “in vivo” term. However the photothemal therapy was perform on cells. I think it should be add “in vitro” term before the “photothermal therapy”.
  2. Why did authors used polyacrylic acid to stabilize the Fe-Au nanoparticles?
  3. Authors should give the discussion and comparison with similar nanoparticles which have reported.

Author Response

Dear Reviewer #5,

First of all, we would like to sincerely thank you for a scrupulous evaluation of the revised version of our manuscript and for the valuable comments. Please find below the detailed answers to the questions mentioned in your review (the Reviewer’s remarks are presented in bold, and our answers follow in plain text, quotations from the manuscript are shown in italic).

  1. The title includes the “in vivo” term. However the photothemal therapy was perform on cells. I think it should be add “in vitro” term before the “photothermal therapy”.

Corrected.

  1. Why did authors used polyacrylic acid to stabilize the Fe-Au nanoparticles?

To emphasize exceptional properties of polyacrylic acid as a stabilizer of nanoparticle colloidal solution, we added the following statement in the Results and Discussion section:

“PAA is a polymer, which bears all qualities suited for biomedical applications, such as biocompatibility, nontoxic nature, and biodegradability”.

  1. Authors should give the discussion and comparison with similar nanoparticles which have reported.

Corrected.